# AANAT1 regulates insect midgut detoxification through the ROS/CncC pathway
Tian Zeng, Fei-yue Teng, Hui Wei, Yong-yue Lu ⓘ, Yi-juan Xu ⓘ ✉ & Yi-xiang Qi ⓘ ✉

Insecticide resistance has been a problem in both the agricultural pests and vectors. Revealing the detoxification mechanisms may help to better manage insect pests. Here, we showed that arylalkylamine N-acetyltransferase 1 (*AANAT1*) regulates intestinal detoxification process through modulation of reactive oxygen species (ROS)-activated transcription factors cap"n"collar isoform-C (CncC): muscle aponeurosis fibromatosis (Maf) pathway in both the oriental fruit fly, *Bactrocera dorsalis*, and the arbovirus vector, *Aedes aegypti*. Knockout/knockdown of *AANAT1* led to accumulation of biogenic amines, which induced a decreased in the gut ROS level. The reduced midgut ROS levels resulted in decreased expression of *CncC* and *Maf*, leading to lower expression level of detoxification genes. *AANAT1* knockout/knockdown insects were more susceptible to insecticide treatments. Our study reveals that normal functionality of AANAT1 is important for the regulation of gut detoxification pathways, providing insights into the mechanism underlying the gut defense against xenobiotics in metazoans.

The intestinal tract of most metazoans is responsible for not only nutrient digestion and absorption but also defense against xenobiotics[1–3]. When insects ingest toxic compounds, the midgut produces a variety of detoxification enzymes that play important roles in adapting to an environment altered by endo- and exogenic compounds, such as host plant defense molecules (allelochemicals) and pesticides[4]. Among these enzymes, cytochrome P450 monooxygenases (CYPs or P450s)[5], carboxylesterases (CarEs)[6], glutathione S-transferases (GSTs)[2], and uridine diphosphate-glycosyltransferases (UGTs)[7] are vital for detoxification of toxins in insects. Earlier studies have shown that insects coopted a xenobiotic stress response pathway involving reactive oxygen species (ROS), cap"n"collar isoform-C (CncC) and its heterodimeric partner, muscle aponeurosis fibromatosis (Maf), which act as transcription factors for overexpression of enzymes and transporters involved in insecticide resistance[8,9].

Arylalkylamine N-acetyltransferases (AANATs) have been identified in various organisms ranging from prokaryotes to eukaryotes and belong to a large superfamily of GCN5-related acetyl-transferases that catalyze transacetylation between acetyl-CoA and arylalkylamines[10]. AANAT1 plays critical roles in cuticle tanning, melatonin synthesis and sleep homeostasis in insects[10–13]. In *Drosophila*, mutations in *AANAT1* correspond to *speck*, characterized by a darkly pigmented region at the wing hinge[14]. While AANAT1 is reported to acetylate and inactivate monoamines (such as serotonin, dopamine, octopamine, and tyramine) in vitro[13], its physiological role of in vivo remains largely unknown. Our previous study revealed that

serotonin modulates *B. dorsalis* gut ROS level[15]. Considering *AANAT1* is highly expressed in insect gut[13,16], and ROS is involved in the transcription of detoxification-related genes[8], we propose that AANAT1 plays a regulatory roles in gut to modulate insect detoxification process.

Herbivorous insects are the big enemies of crops, and insect vectors threaten public health by transmitting infectious diseases. So far, Chemical control remains the most important and widely used strategy against these insect pests around the world. However, studies have shown that these insects develop resistance to a range of insecticide classes. Therefore, studying the detoxification mechanisms of these pests may help to better manage them. In this study, we demonstrate that AANAT1 plays an important role in midgut detoxification via modulation of ROS levels in a major agricultural pest *Bactrocera dorsalis* and the mosquito *Aedes aegypti*.

## Results

### AANAT1 modulates midgut detoxification in *B. dorsalis*

To investigate the physiological role of AANAT1 in *B. dorsalis*, we obtained a *BdorAANAT1* knockout strain with a 10-bp deletion, designated *AANAT1*[10/10] (Fig. 1a)[13]. Knockout of *BdorAANAT1* was confirmed at both genomic and protein levels (Fig. 1a, b). Western blot analysis using Bdor-AANAT1 antibody revealed a protein band at the predicted size (27 kDa) in wild-type (WT) strain but not mutant groups (Fig. 1b). Next, we investigated the differences in gut detoxification processes between WT and

Department of Entomology, College of Plant Protection, South China Agricultural University, Guangzhou, China.
✉e-mail: xuyijuan@scau.edu.cn; qiyixiang@scau.edu.cn

**Fig. 1 | AANAT1 gene knockout in *B. dorsalis* increased susceptibility to insecticide. a** A *Bdor-AANAT1* knockout strain, with 10-bp deletions, was obtained by CRISPR/Cas9. **b** Confirming *Bdor-AANAT1* knockout efficiency at the protein level by western blot analysis. The samples were extracted from adult midguts. **c** Susceptibility of the WT and *AANAT1^{10/10}* strain flies' responses to diverse insecticides by feeding. $LC_{50}$, lethal concentration that kills 50% of *B. dorsalis* adults; 95% FL, 95% fiducial limits of $LC_{50}$. Increased folds in susceptibility were calculated as $LC_{50}$ of WT/$LC_{50}$ of *AANAT1^{10/10}*. $LC_{50}$ values were considered significantly different if their 95% FL did not overlap, and asterisks indicate statistical significance of $LC_{50}$ between the two strains.

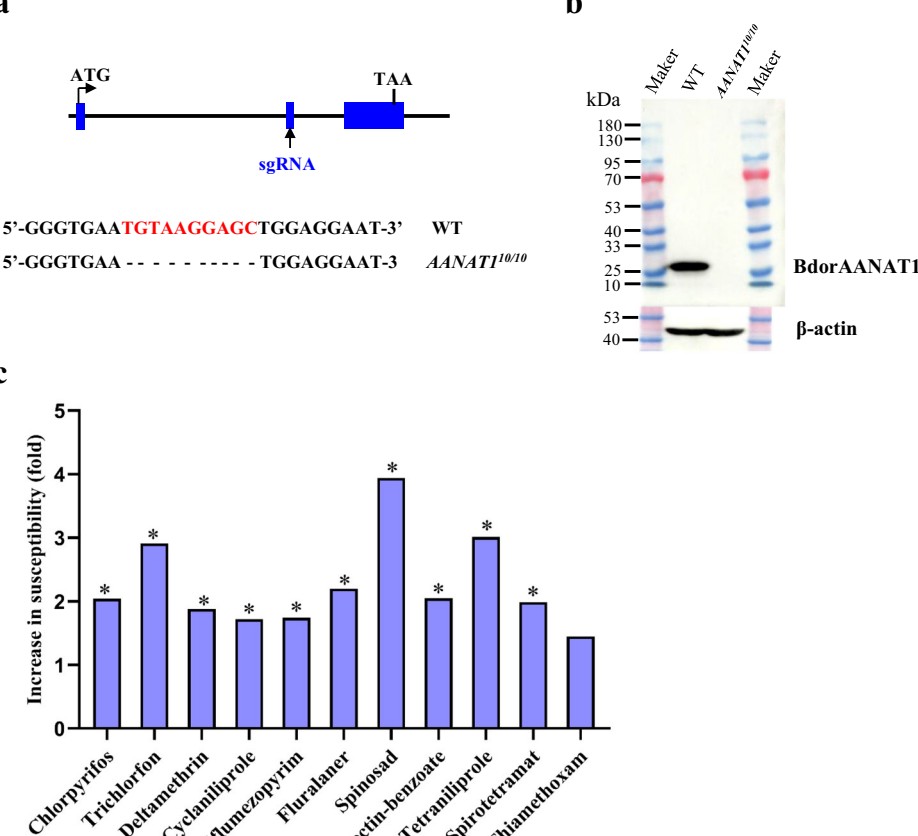

*AANAT1^{10/10}* strains. The responses of adults from the *BdorAANAT1* knockout and WT background strains to a series of concentrations of insecticide by feeding were determined. Compared to WT, significant decreases in $LC_{50}$ values of *AANAT1^{10/10}* strain were observed for chlorpyrifos (2.04 fold), trichlorfon (2.91 fold), deltamethrin (1.88 fold), cyclaniliprole (1.72 fold), triflumezopyrim (1.74 fold), fluralaner (2.20 fold), spinosad (3.94 fold), emamectin-benzoate (2.05 fold), tetraniliprole (3.01 fold), and spirotetramat (1.99 fold) (Supplementary Table 2 and Fig. 1c). However, there was no significant difference in the $LC_{50}$ value of thiamethoxam between the two strains (Supplementary Table 2 and Fig. 1c).

Then, to determine the expression patterns of AANAT1 in the intestinal tract of *B. dorsalis*, RT-qPCR was conducted. *BdorAANAT1* transcripts were mainly detected in the midgut region (Fig. 2a). Expression of BdorAANAT1 protein in midgut epithelia of the WT group was confirmed via immunohistochemistry (Fig. 2b). Notably, BdorAANAT1 immunoreactivity was not detected in midgut of the mutant strains (Fig. 2b). We detected weak fluorescence signals of BdorAANAT1-positive cells in Malpighian tubules (Supplementary Fig. 1a, b). No BdorAANAT1 immunoreactivity was detected in the fat body (Supplementary Fig. 1c, d). Using RNA sequencing (RNA-seq), expression profiles of midguts of WT and *AANAT1^{10/10}* strains were compared. Principal component analysis (PCA) revealed significant differences between the *AANAT1^{10/10}* strain and WT strain (Fig. 2c), for instance, in a group of genes involved in detoxification (Fig. 2d). Compared with WT, eight GSTs, five CYPs, and four UGTs were downregulated. Differential expression of these genes was confirmed via RT-qPCR (Fig. 2e–t). Compared with WT, activities of GST and P450 were reduced in the *AANAT1^{10/10}* strain (Fig. 2u, v).

## BdorAANAT1 modulates midgut detoxification through ROS production

Previous studies suggest that ROS are implicated in the regulation of genes involved in detoxification in insects[8]. Accordingly, ROS activity in the

midgut of WT and *AANAT1^{10/10}* strains was examined in this study. Compared with WT, lower hydrogen peroxide ($H_2O_2$) production and weaker ROS signals were observed in the midgut of *BdorAANAT1* knockout flies (Fig. 3a, b). Next, WT flies were fed the antioxidant vitamin C and midgut activity of detoxification enzymes monitored. Treatment with vitamin C effectively inhibited midgut $H_2O_2$ production (Fig. 3c). In addition, GST and P450 activities were markedly suppressed following vitamin C treatment (Fig. 3d, e). Expression levels of the randomly selected genes *GstE4*, *GstE7*, *GstE9*, *Cyp6g2*, *UGT36-D1*, and *UGT49-C1* in the midgut of vitamin C-treated flies were much lower than the corresponding values in the control group (Fig. 3f–k). Compared to control, significant decreases in $LC_{50}$ value of vitamin C-treated flies was observed for trichlorfon (5.07 fold) (Fig. 3l).

To confirm these findings, knockdown of *BdorAANAT1* was performed via RNA interference (RNAi). Flies treated with GFP double-stranded RNA (dsRNA) were used as the control group. The *AANAT1* transcript level in *B. dorsalis* midgut was inhibited at 24–72 h post RNAi. The strongest knockdown effect (approximately 65%) was observed at 48-h post RNAi treatment (Supplementary Fig. 2a). Compared to GFP dsRNA-treated flies, *BdorAANAT1* dsRNA-treated flies showed decreased $H_2O_2$ levels in the midgut (Supplementary Fig. 2b). Midgut GST and P450 activities were decreased in *BdorAANAT1* knockdown flies (Supplementary Fig. 2c, d). Additionally, expression of detoxification genes, such as *GstE9*, *Cyp6g2*, *UGT36-D1*, and *UGT49-C1* was significantly lower in the midgut of ds*BdorAANAT1*-treated than control group (Supplementary Fig. 2e–h). These results are consistent with data from *BdorAANAT1* knockout experiments. To detect the role of AANAT1 in insecticide resistance, flies of a resistant strain were injected with dsRNA to silence the expression of *BdorAANAT1* gene. $LC_{50}$ values of the *BdorAANAT1* knockdown flies decreased significantly for trichlorphon (1.72-fold) when compared with the control (Supplementary Fig. 2i). However, expression levels of

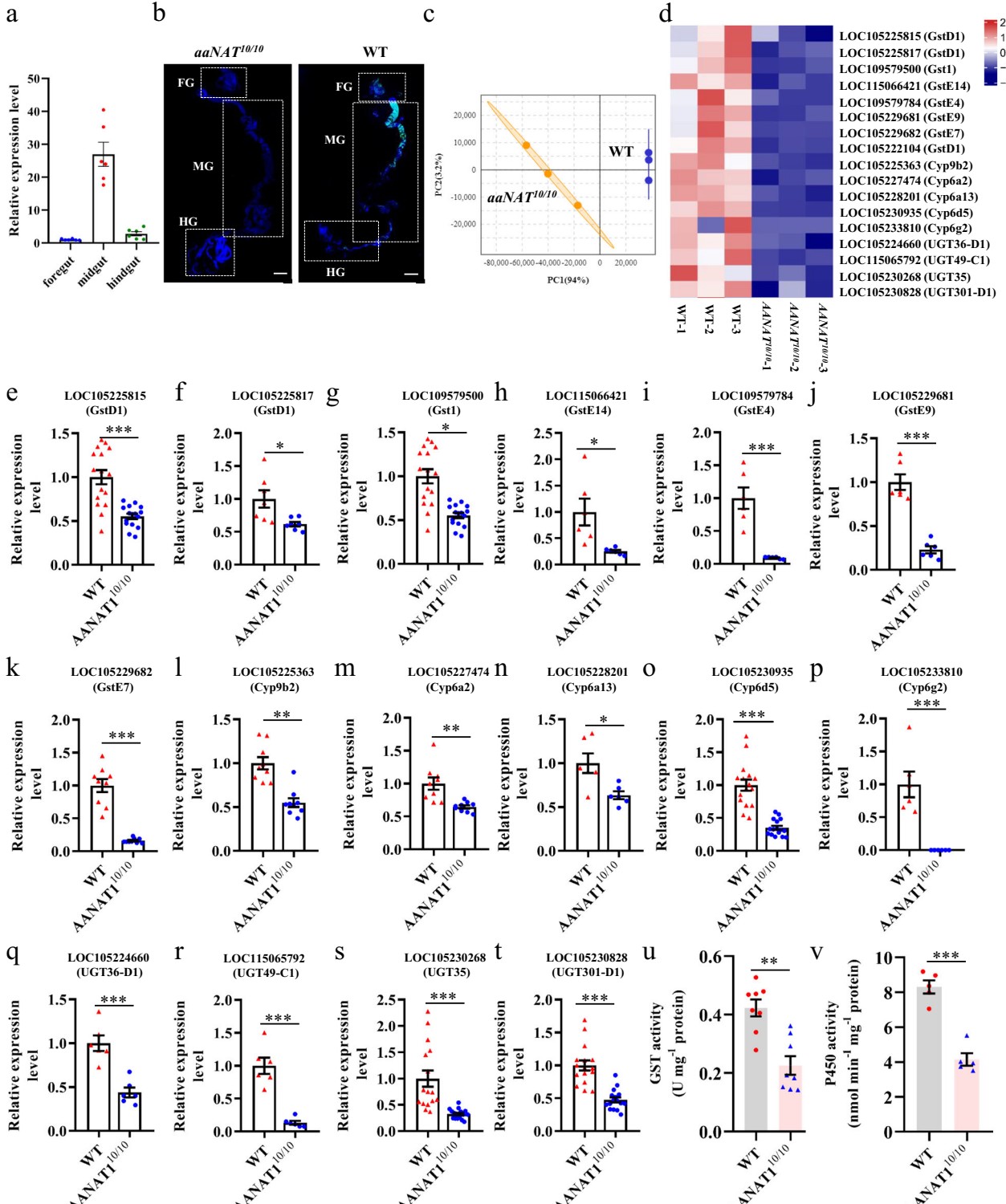

**Fig. 2 | BdorAANAT1 modulates midgut detoxification gene expression.**
**a** Relative expression level of *BdorAANAT1* in different gut region of *B. dorsalis*.
**b** Immunostaining of *B. dorsalis* gut using BdorAANAT1 antibody. FG, foregut, MG, midgut, HG, hindgut. The scale bar represents 1000 μm. **c** Principal component analysis of midgut RNA-seq from WT and *AANAT1^{10/10}* strains. R package gmodels was used in this experiment. **d** Heat map showing the relative expression levels of genes involved in detoxification in the midgut of *B. dorsalis*. The map is plotted based

on Log2-transformed FPKM values, each bar or column corresponds to the relative gene expression level of gene in one sample, with warmer colors representing higher relative gene expression levels. **e–t** qPCR validation of differentially expressed genes in the midgut of WT and *AANAT1^{10/10}* strains. **u** GST activity in the midgut of *B. dorsalis*. **v** P450 activities were determined using p-nitroanisole-O-demethylase (PNOD) as the substrate. Student's *t* test was performed for **e–v**. Error bars indicate ± s.e.m., \*\*\**p* < 0.001, \*\**p* < 0.01, \**p* < 0.05.

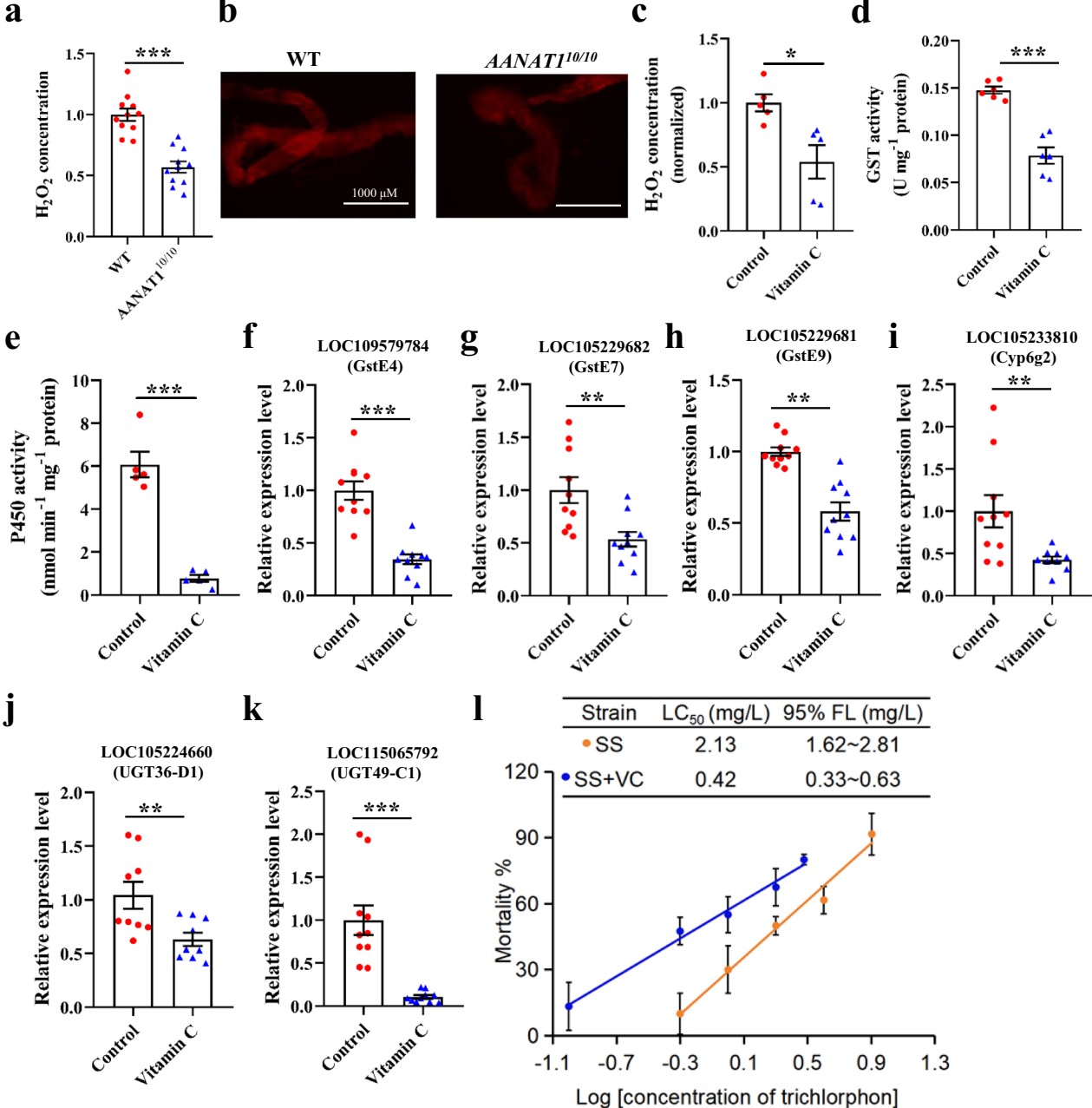

**Fig. 3 | Decreasing ROS level by vitamin C treatment affected midgut detoxification in *B. dorsalis*.** ROS activity of the midgut from WT and *AANAT1^10/10* strain flies was measured by $H_2O_2$ assay (**a**) and dihydroethidium (DHE) (**b**). Scale bars in **b** represent 1000 μm. Data of $H_2O_2$ level are normalized to WT. **c** $H_2O_2$ concentration was determined in the midgut of *B. dorsalis* after vitamin C treatment. Data of $H_2O_2$ level are normalized to controls. **d**, **e** GST and P450 activities were determined in the midgut of *B. dorsalis* after vitamin C treatment.
**f–k** Transcriptional responses of *LOC109579784 (GstE4)*, *LOC105229682 (GstE7)*,

*LOC105229681 (GstE9)*, *LOC105233810 (Cyp6g2)*, *LOC105224660 (UGT36-D1)*, and *LOC115065792 (UGT49-C1)* to vitamin C treatment. **l** Effect of vitamin C treatment on flies to trichlorphon. SS, sensitive strain, VC, vitamin C. Ten-day old adults mixed with the same number of males and females were treated with 100 mg/mL vitamin C for 24 h. Then, flies were treated with trichlorphon or their midguts were dissected on ice. Student's *t* test was performed for **a**, **c**, **d–k**. Error bars indicate ±s.e.m., ***$p < 0.001$, **$p < 0.01$, *$p < 0.05$.

*BdorAANAT1* in the intestine of sensitive (SS) and resistant strains (RS) to trichlorphon were not significantly different (Supplementary Fig. 2j).

Our previous study revealed that BdorAANAT1 modulates ovary development[13]. To detect whether the knockout of this gene affect the intestinal development of flies, we compared their gut sizes. Knockout of *BdorAANAT1* did not significantly affect the length of the foregut, midgut, and hindgut (Supplementary Fig. 3a). Compared with WT, the midgut width of *AANAT1^10/10* mutants did not change significantly (Supplementary Fig. 3b). Furthermore, no significant difference was observed in the number of phospho-histone H3 (PH3)-positive cells and 4',6-diamidino-2-

phenylindole (DAPI)-positive cells in the midgut between WT and *AANAT1^10/10* strain (Supplementary Fig. 3c–f). These results indicate that cell proliferation was not significantly affected in the midgut of *BdorAANAT1* knockout flies. Therefore, the increased sensitive to insecticides of *AANAT1^10/10* strain is not due to gut development.

## BdorAANAT1 regulates detoxification gene expression via a ROS-activated CncC: Maf pathway

Studies reveal that *CncC* forms a heterodimer with Maf to regulate the transcription of detoxification genes in a range of insects, including *B.*

**Fig. 4 | ROS activated CncC: Maf pathway is involved in BdorAANAT1 regulated midgut detoxification process.** mRNA level of *Bdor-CncC*, *Bdor-Maf*, and *Bdor-Keap1* in the midguts of *Bdor-AANAT1* knockout (**a**), and knockdown (**b**) flies. **c** The effect of vitamin C treatment on the midgut expression of *Bdor-CncC*, *Bdor-Maf*, and *Bdor-Keap1*. **d** Effect of overexpression of transcription factor *BdCncC* on the promoter activity of P450 and GST genes. Relative fluorescence (RLU) activity = Firefly luciferase activity/Renilla luciferase activity. **e** Midgut *Bdor-CncC* silencing efficiency at 72 h post RNAi. **f, g** Regulation of both GST and P450 activities in the midguts of *CncC*-silencing flies. **h–v** Transcriptional responses of detoxification genes to *Bdor-CncC* knockdown. Student's *t* test was performed. Error bars indicate ±s.e.m., ****p* < 0.001, ***p* < 0.01, **p* < 0.05.

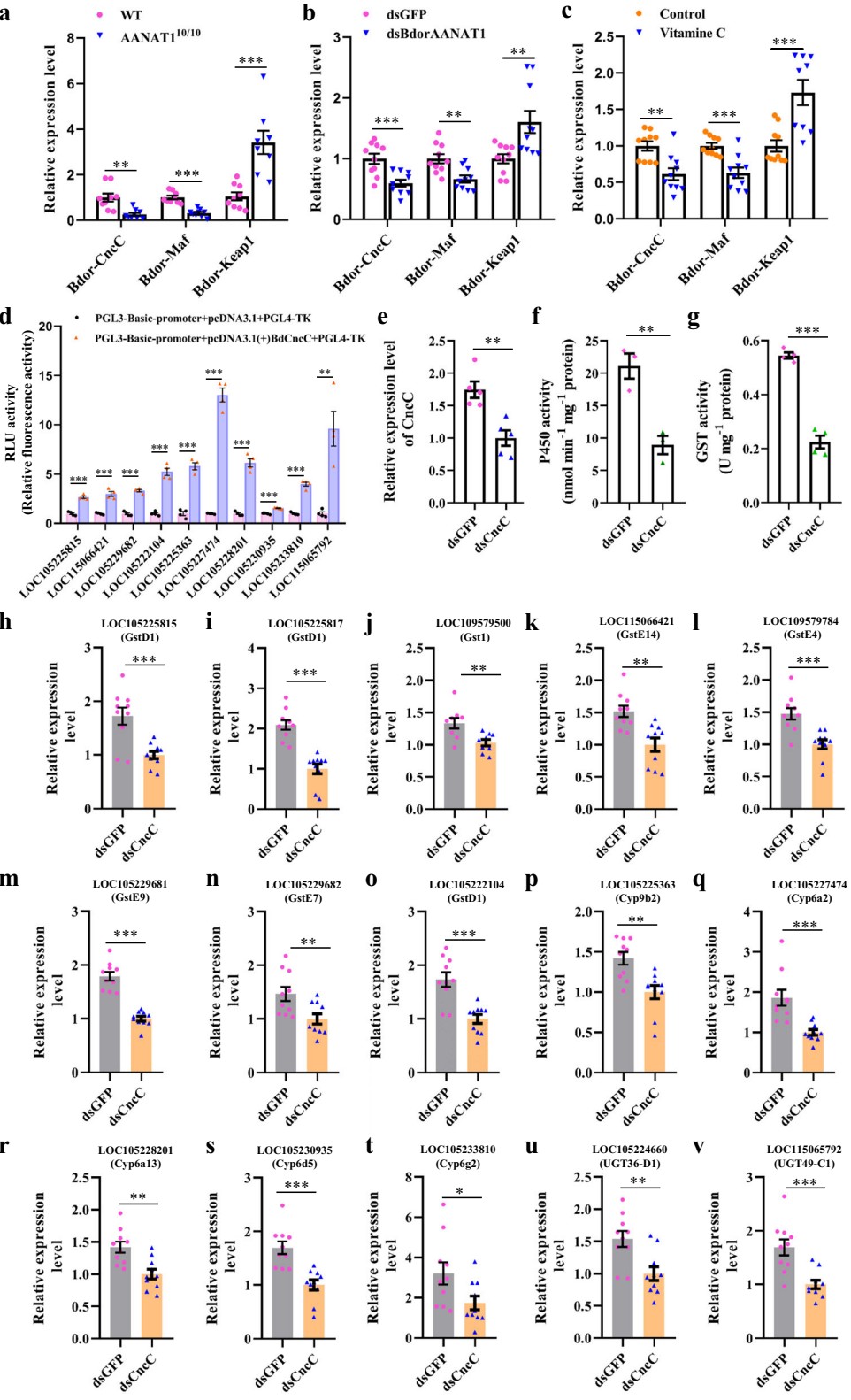

*dorsalis*[8,17]. Compared with controls, transcript levels of *CncC* and *Maf* in the midgut of *BdorAANAT1* knockout/knockdown flies were decreased (Fig. 4a, b). However, the mRNA level of *Keap1* that sequesters CncC under normal conditions was increased in the midgut of *BdorAANAT1* knockout/knockdown flies (Fig. 4a, b). In accordance with these results, vitamin C treatment inhibited the expression of *CncC* and *Maf*, concomitant with an increase in *Keap1* (Fig. 4c). Moreover, predicted results showed the presence of the CncC–Maf

binding site in all differentially expressed detoxification genes (Supplementary Fig. 4). Furthermore, using a dual-luciferase reporter gene assay, we confirmed that overexpression of *BdCncC* significantly enhances the promoter activity of the 10 targeted P450 and GST genes by 1.55–13.02-fold compared to the pcDNA3.1(+) control (Fig. 4d). Silencing the expression of *CncC* in flies of sensitive strain by RNAi decreased midgut GST and P450 activities (Fig. 4e–g). The expression level of detoxification genes was significantly lower in the midgut of

**Fig. 5 | BdorAANAT1 controls ROS production via monoamines in the midguts of *B. dorsalis*.** Midgut monoamines including serotonin (**a**), dopamine (**b**), tyramine (**c**), and melatonin (**d**) was determined using HPLC-MS. **e** Amine mix including serotonin, dopamine, and tyramine inhibited $H_2O_2$ level in a dose- dependent manner. **f** The effect of 10 μM serotonin, dopamine, and tyramine on the $H_2O_2$ level of WT strain. **g** The effect of 10 μM melatonin on the $H_2O_2$ level of *AANAT1^10/10* strain. Student's *t* test was performed for **a**–**f**. Error bars indicate ±s.e.m., ***$p < 0.001$, **$p < 0.01$, *$p < 0.05$.

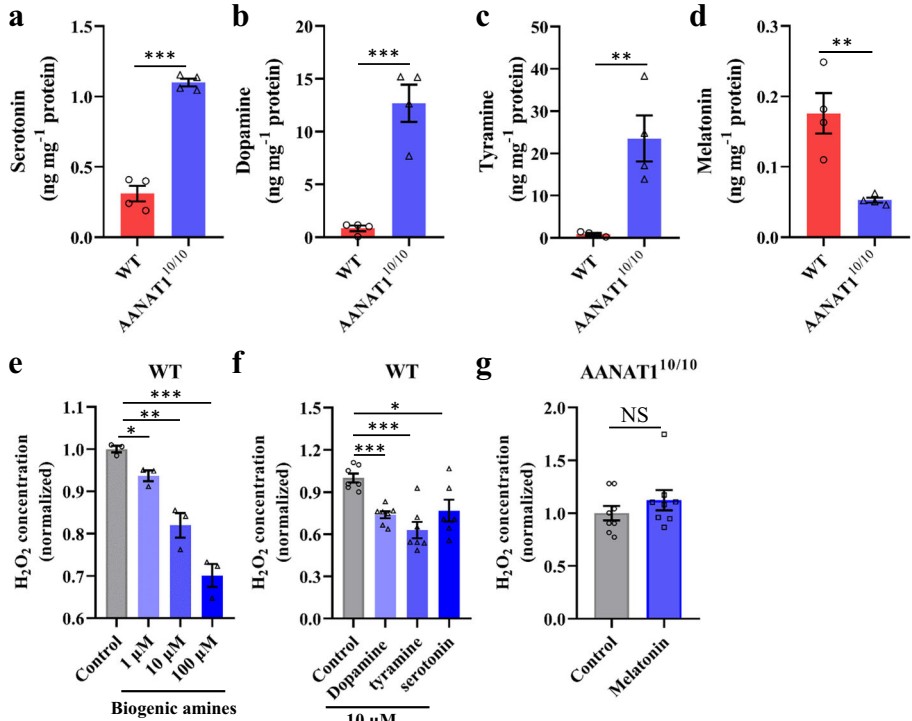

ds*CnCc*-treated than control group (Fig. 4h–v). Together, these findings indicate that AANAT1 regulates detoxification gene expression via the ROS-activated CncC: Maf pathway.

In insects, the nicotinamide adenine dinucleotide phosphate (NADP)H oxidase family, notably Duox and Nox, are primarily involved in generation of ROS[18,19]. However, *Duox* and *Nox* mRNA levels were comparable between WT and *AANAT1^10/10* strains (Supplementary Fig. 5), indicating that AANAT1 may regulate other ROS production-related processes. The above results suggest that *BdorAANAT1* knockout induces a decrease in the midgut ROS level, resulting in reduced detoxification gene transcription.

### BdorAANAT1 controls ROS production through monoamines

Earlier in vitro studies have shown that monoamines, including serotonin, dopamine, octopamine, and tyramine are substrates for BdorAANAT1[13]. Our previous study revealed that serotonin modulates *B. dorsalis* gut microbiome homeostasis via regulation of ROS[15]. Moreover, in vertebrates, AANAT is the first and rate-limiting enzyme for biosynthesis of melatonin, which is a highly effective antioxidant agent as well as free radical scavenger[20]. Here, we addressed the issue of whether these amines and melatonin are regulated by AANAT1 in insects. To this end, high performance liquid chromatography—mass spectrometry (HPLC-MS) was employed to evaluate the levels of dopamine, serotonin, octopamine, tyramine, and melatonin in midgut regions of *AANAT1^10/10* and control flies. The midgut of the *AANAT1^10/10* strain showed a robust increase in serotonin, dopamine and tyramine levels compared to its WT counterpart (Fig. 5a–c). However, octopamine was undetectable in the midgut of both WT and *AANAT1^10/10* strains. The melatonin level was significantly reduced in the midgut of *BdorAANAT1* knockout flies (Fig. 5d). Next, we determined the effects of these amines and melatonin on midgut ROS activity in *B. dorsalis*. The biogenic amine mix (including serotonin, dopamine, and tyramine) suppressed the midgut $H_2O_2$ level in WT flies in a dose- dependent manner (Fig. 5e). Moreover, all the three tested amines inhibited midgut $H_2O_2$ level when they work alone (Fig. 5f). In contrast, 10 μM melatonin had no significant effect on the midgut $H_2O_2$ content of the *AANAT1^10/10* group (Fig. 5g). These data indicate that BdorAANAT1 regulates biogenic amines level, which decrease the ROS level.

### AANAT1 affects midgut detoxification through ROS/CncC pathway in *A. aegypti*

The above findings indicate AANAT1 is a potential target to manage pest. In consideration of the difficulties in controlling *A. aegypti*, a key vector for human diseases, including dengue, yellow fever, chikungunya and Zika, we were determined to detect the detoxification role of AANAT1 in the midgut of this mosquito. *AANAT1* of *A. aegypti* was previously identified[21]. The *AANAT1* gene was knocked down via thoracic microinjection of dsRNA into *A. aegypti*. Compared with controls, midgut mRNA level of *AaAANAT1* was significantly down-regulated post dsRNA injection (Fig. 6a).The strongest knockdown effect (approximately 75.6%) was observed at 72-h post RNAi treatment (Fig. 6a). Consistent with the findings in *B. dorsalis*, knockdown of *AaAANAT1* led to decreased $H_2O_2$ levels (Fig. 6b). Transcripts abundance of *Aa-CncC* and *Aa-Maf* were decreased, while mRNA level of *Aa-Keap1* was increased in the midguts of *AaAANAT1*-RNAi mosquitoes compared with those in *GFP*-RNAi controls (Fig. 6c). Moreover, enzyme activities of both P450 and GST were significantly decreased in the gut of *AaAANAT1*-silenced mosquitoes (Fig. 6d, e). Then, the responses of *AaAANAT1*-RNAi mosquitoes and controls to a series of concentrations of cypermethrin by feeding were determined. The dose-response curves are presented in Fig. 6f. The $LC_{50}$ value was 4.228 mg/L for control strain and 2.090 mg/L for *AaAANAT1*-RNAi mosquitoes. Compared with GFP-RNAi controls, $LC_{50}$ value was significantly decreased (2.02-fold) in *AaAANAT1*-RNAi strain. These findings suggest that AANAT1 affects insect midgut detoxification by ROS-activated CncC pathway (Fig. 6g).

### Discussion

There is a constant battle between humans and insects. Although, some insects are beneficial, others are disease vectors or pests. Humans have developed methods to control them, such as pesticide. However, insects have developed resistance to almost all classes of chemicals introduced to control them. Therefore, revealing the detoxification mechanisms may help to better manage pests. In this study, AANAT1 has been identified as a critical factor in regulation of detoxification in the intestine. This regulatory activity is dependent on biogenic amines that influence ROS level.

Chemical insecticides can be effectively metabolized by insect detoxification enzymes via three stages: phase I (oxidation, hydrolysis and

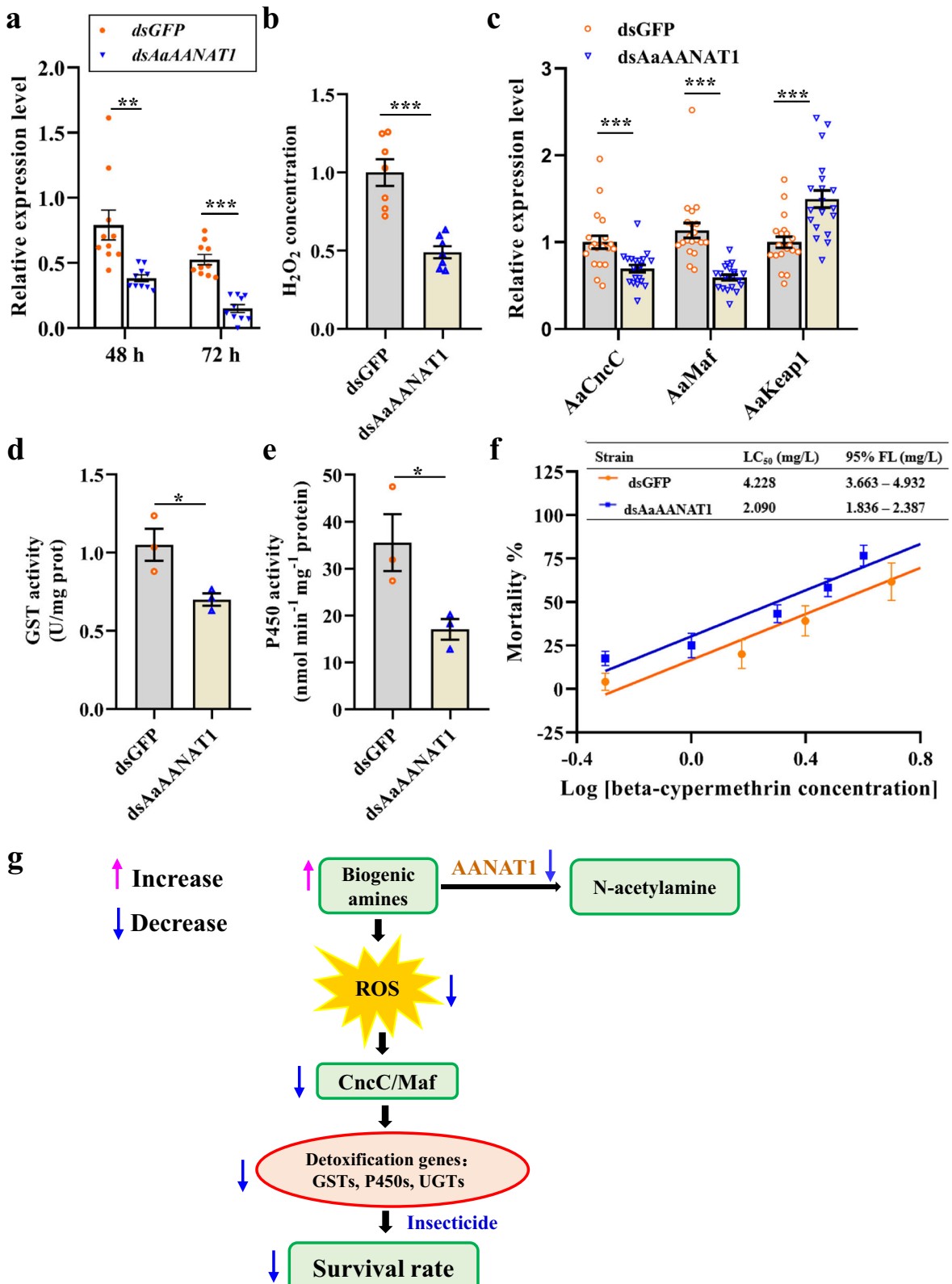

**Fig. 6 | AANAT1 is involved in midgut detoxification via ROS/CncC pathway in *A. aegypti*. a** Midgut *AANAT1* silencing efficiency determination in *A. aegypti* by real-time PCR. **b** Effects of *Aa-AANAT1* silencing on midgut $H_2O_2$ concentration. Data of $H_2O_2$ level were normalized to controls. **c** Transcripts level of *Aa-CncC*, *Aa-Maf*, and *Aa-Keap1* in the midguts of *Aa-AANAT1* knockdown mosquitoes. **d**, **e** GST and P450 activities were determined in the midgut of *A. aegypti* after dsRNA treatment. **f** Effect of *Aa-AANAT1* silencing on the susceptibility of mosquitos to beta-cypermethrin. $LC_{50}$ lethal concentration that kills 50% of *B. dorsalis* adults; 95%

FL, 95% fiducial limits of $LC_{50}$. **g** Schematic representation of AANAT1 modulates midgut detoxification process. AANAT1 regulates the concentration of gut biogenic amines, which reduce the ROS level. The decreased ROS level inhibited detoxification gene transcription via Cncc:Maf pathway. Decreased detoxification enzyme activity reduced insect survival rate when they are treated with insecticide. Student's *t* test was performed for **a–e**. Error bars indicate ±s.e.m., ***$p < 0.001$, **$p < 0.01$, *$p < 0.05$. $LC_{50}$ values were considered significantly different if their fiducial limits did not overlap.

reduction), phase II (conjugation) and phase III (excretion)[8]. Constitutive or induced overexpression of genes coding for enzymes involved in detoxification (P450s) or conjugation (GST, UGT) has been reported in insects that develop resistance to most classes of insecticides[7,22,23]. Studies have demonstrated that the transcription factor CncC, induced by ROS, is a central regulator of insect response to xenobiotics that acts by inducing the expression of detoxification genes, including P450, GST, and UGT[7,24]. Under normal conditions, CncC heterodimerizes with Keap1 and remains in the cytoplasm anchored to actin filaments. Under xenobiotic stress conditions, ROS or other molecules promote dissociation of CncC and Keap1, facilitating translocation of CncC to the nucleus and its heterodimerization with Maf. The heterodimer binds CncC/Maf response elements located in the promoter regions of xenobiotic response genes and stimulates their expression[8]. In this study, putative antioxidant response elements were identified in the promoter regions of detoxification genes affected by *AANAT1* knockout, indicating that ROS-related signaling pathways may contribute to the regulation of these genes. We further confirmed that overexpression of *BdCncC* significantly enhances the promoter activity of targeted P450 and GST genes. Due to reduced ROS levels in the gut of *AANAT1* knockout flies, mRNA levels of *Cncc* and *Maf* declined, while expression level of *Keap1* was increased, leading to reduced transcription of detoxification enzymes. Our results showed that the *AANAT1* knockout flies and mosquitos were more susceptible to insecticide. Moreover, in resistant strain of *B. dorsalis*, knockdown of *AANAT1* by RNAi resulted in elevated susceptibility to trichlorphon. Therefore, this protein appears to play a key role in detoxification of insect midgut. Expression level of *AANAT1* was not significantly increased in the resistant strain of *B. dorsalis*, indicating that detoxification is very complicated and other factors affect this process. Moreover, AANAT1 is critical for the metabolism of amines, including some neurotransmitters, which modulate insect physiological processes and behaviors. Therefore, constitutive expression of AANAT1 is important for insect survival.

Production of ROS, including $H_2O_2$, hydroxyl radicals ($-OH$) and superoxide anions ($O^{2-}$), is ubiquitous in animals in response to chemical stress or pathogen attack[25,26]. Insect cellular ROS are generated by Duox and Nox, two members of the nicotinamide adenine dinucleotide phosphate (NADPH) oxidase family[18,19]. ROS can be also formed as a byproduct of endogenous metabolism. To alleviate the oxidative stress caused by ROS, insects have developed many antioxidant enzyme systems, including superoxide dismutase, peroxidase and catalase[27]. The transcript abundance of *Duox* and *Nox* was not affected by *AANAT1* knockout, suggesting that *AANAT1* may regulate other ROS production-related processes, such as enzyme activity, ROS scavenging or oxidative metabolism. Biogenic amines, including serotonin, dopamine, and tyramine, accumulate in *BdorAANAT1* knockout flies, since they serve as substrates of the AANAT1 enzyme[13,28]. Moreover, in insects, monoamine oxidase (MAO) activity is limited, which is critical in the inactivation of catecholamines and other aromatic amines in mammals[29]. Although a previous study by our group revealed that serotonin inhibits ROS production by regulating *Duox* gene expression[15], the mRNA level of *Duox* was not affected by *AANAT1* knockout. In consideration of the diverse substrates of AANAT1 and their corresponding products, it is proposed that other compounds could regulate *Duox* expression in an opposite manner. Moreover, compared with serotonin, accumulation of dopamine and tyramine was markedly higher in *AANAT1* knockout flies. The precise mechanisms by which biogenic amines contribute to control of ROS levels in the midgut are not clearly understood. Multiple mechanisms may be involved in amines-mediated regulation of ROS levels. First, amines (such as dopamine) may play a role in ROS scavenging. Dopamine is known to possess the ortho-dihydroxyphenyl (catechol) functional group, which provides the ability to scavenge ROS in a dose-dependent manner[30,31]. Second, biogenic amines could regulate ROS production via interactions with their receptors. In mammals, dopamine D1-like and D5 receptors have been shown to inhibit ROS production[32,33].

In conclusion, the results of this study reveal an important role of AANAT1 in midgut detoxification. Therefore, normal functionality of AANAT1 in the midgut is important for insect survival when they ingest toxic compounds. Our findings may provide new targets for pest management. AANATs have been identified in various organisms, including vertebrates and invertebrates. Whether the detoxification role of AANAT in the intestinal tract is conserved in other organisms will need further investigation.

## Materials and methods
### Insect strains and rearing
Wild-type susceptible strain (SS) was originally collected from a carambola (*Averrhoa carambola*) orchard located in Guangzhou, Guangdong Province, in April 2008[34]. A resistant strain (RS) of *B. dorsalis* was obtained by selection after adult exposure to a trichlorphon treated surface over the course of 40 generations. Trichlorphon was diluted with acetone to $LC_{50}$ for each generation and evenly coated onto the inside of a 250-mL conical flask by shaking. After the acetone had volatilized, 5-day-old fly pairs were placed in the flask for 24 h. The surviving flies were selected for breeding of the next generation[34]. The *BdorAANAT1*$^{10/10}$ knockout strain based on WT was obtained using the CRISPR-Cas9 genome editing tool[13].

Theses strains were maintained in the laboratory under the following conditions: 25 ± 1 °C, 16:8 h light: dark cycle, 70–80% relative humidity (RH). Larvae were reared on an artificial diet comprising 150 g corn flour, 150 g banana, 0.6 g sodium benzoate, 30 g yeast, 30 g sucrose, 30 g paper towel, 1.2 mL hydrochloric acid and 300 mL water. Adults were fed an artificial diet consisting of yeast extract and dry sugar mixed at a 1:1 ratio (w/w) and housed in transparent plastic cages.

*Aedes aegypti* was reared at 26 °C and 80% relative humidity under a 16:8 h light: dark photoperiod. Mosquito larvae were raised on rat chow. Male and female adult mosquitoes were maintained in a cage with unlimited access to cotton balls moistened with 8% sucrose solution. Ten-day-old virgin females without blood feeding were used in the experiments.

### Western blot and immunostaining analysis
Details of the procedure used to obtain recombinant BdorAANAT1 protein are described in a previous report by our group[13]. Purified protein was used as an antigen for immunization in rabbits. The antibody specific for BdorAANAT1 was purified from antiserum provided by GenScript Co., Ltd. (Nanjing, China).

Twenty midgut sections from 15-day-old adult flies were lysed in ice-cold RIPA lysis buffer I (Sangon Biotech, Shanghai, China) containing protease inhibitors and phenylmethyl sulfonylfluoride (Sangon Biotech). Homogenates were centrifuged for 5 min at 4 °C and $12,000 \times g$ and the supernatant collected. Protein concentrations were measured using the Bradford assay. Samples were diluted in 5× Protein SDS PAGE Loading Buffer (Sangon Biotech) and boiled for 7 min, followed by centrifugation for 5 min at 4 °C and $12,000 \times g$. The supernatant was separated in a denaturing polyacrylamide gel and transferred to PVDF membrane. After blocking (5% nonfat dry milk in Tris-buffered saline containing 0.1% Tween 20; pH 7.4) and washing, membranes were incubated overnight with primary antibodies against BdorAANAT1 (1:1000) and beta-actin (1:1000), followed by secondary horseradish peroxidase–conjugated goat anti-rabbit IgG (Sangon Biotech) diluted 1:1000 in Tris-buffered saline with Tween-20. Membranes were rinsed five times with wash buffer and incubated with ECL western blot substrate (Sangon Biotech).

For immunostaining analysis, the guts, fat body, and Malpighian tubules of WT and *BdorAANAT1* knockout adult flies were dissected in phosphate-buffered saline (PBS) and immediately placed in 4% paraformaldehyde at room temperature for 2 h. After fixation, samples were washed rapidly twice with 0.3% PBST (1.5 mL Triton X-100 dissolved in 498.5 mL PBS) solution and then slowly three times with PBST for 20 min each. Following aspiration of the wash solution, blocking solution (5% goat serum/PBST) was added for 1 h of osmotic blocking at room temperature. Next, samples were incubated with antibodies against BdorAANAT1 (1:100) in blocking solution for 48 h at 4 °C, followed by antibody aspiration and washing steps as above. Samples were treated with goat anti-rabbit IgG

Alexa Fluor Plus 488 (Invitrogen, Carlsbad, CA, USA) and incubated overnight at 4 °C and washed with PBST as above. Nuclei were stained with 4'−6-diamidino-2'-phenylindole (DAPI, Sigma) for 5 min at room temperature and re-washed with PBST as above. Finally, 10 μL fluorescent anti-fade agent was added to the slides, which were coverslipped and sealed with kisse's mounting medium for immediate observation using a Zeiss laser confocal microscope system (LSM 7810 DUO and LSM 7 Live, Carl Zeiss AG, Oberkochen, Germany) or stored in the dark at −20 °C.

### RNA extraction and sequencing

Total RNA was extracted from midguts of 20 flies (aged 10 days after emergence) per biological replicate using TRIzol reagent kit (Invitrogen) according to the manufacturer's protocol. Three biological replicates were analyzed. RNA quality was assessed on an Agilent 2100 Bioanalyzer (Agilent Technologies, Palo Alto, CA, USA) and subjected to RNase-free agarose gel electrophoresis. After extraction of total RNA, eukaryotic mRNA was enriched with oligo(dT) beads while prokaryotic mRNA was enriched by removing rRNA with a Ribo-Zero™ Magnetic Kit (Epicentre, Madison, WI, USA). Enriched mRNA was fragmented using fragmentation buffer and reverse-transcribed into cDNA with random primers. Second-strand cDNA was synthesized using DNA polymerase I, RNase H, dNTPs and buffer, purified with a QiaQuick PCR extraction kit (Qiagen, Venlo, The Netherlands), end-repaired, poly (A) added, and ligated to Illumina sequencing adapters. Ligation products were size-selected via agarose gel electrophoresis, PCR-amplified, and sequenced using Illumina HiSeq2500 (Gene Denovo Biotechnology Co., Guangzhou, China).

Reads obtained from the sequencing machines included raw reads containing adapters or low-quality bases, which affect subsequent assembly and analysis. To obtain high-quality clean reads, further filtration was performed using FASTP (version 0.18.0). Prior to assembly, paired-end raw reads from each cDNA library were processed to remove adapters, low-quality sequences (Q < 20), and reads contaminated with microbes. Clean reads were de novo assembled to produce contigs. An index of the reference genome of *B. dorsalis* was generated to which paired-end clean reads were mapped using HISAT2. 2.4, with '-rna-strandness RF' and other parameters set as default. To evaluate transcript abundance, StringTie software was applied to calculate the normalized gene expression value FPKM. RNA differential expression analysis between two groups was performed with DESeq2 software. Genes/transcripts with a false discovery rate (FDR) below 0.05 and absolute fold change ≥2 were considered differentially expressed. To determine the structure/relationship of samples, principal component analysis (PCA) was performed with the R package 'gmodels' (http://www.r-project.org/).

### Gene expression analysis by reverse-transcription quantitative PCR

To analyze gene expression in the intestinal tract of *B. dorsalis*, the guts of 10 day-old adults (equal number of males and females) were dissected. Experiments were performed independently at least twice, with three to five biological replicates (20 flies per replicate). To analyze gene expression in *A. aegypti*, midguts of ten-day-old females were dissected. Experiments were performed independently at least twice, with five biological replicates, each replicate containing 20 mosquitoes. RNA was extracted using TRIzol reagent (Invitrogen, Carlsbad, CA). The purity of extracted RNA was assessed spectrophotometrically based on the $OD_{260/280}$ ratio, whereby $OD_{260/280}$ values of 1.8–2.0 indicate good quality RNA. RNA integrity was measured via electrophoresis on a formaldehyde agarose gel. RNA aliquots (1 μg) were reverse-transcribed to cDNA using a PrimeScript™ RT reagent Kit with gDNA Eraser (Takara) according to the manufacturer's instructions. Biosynthesized cDNA was used as a template for RT-qPCR, which was conducted on a CFX96™ Real-Time PCR System (Bio-Rad; Hercules, CA, USA) with TB Green Premix Ex Taq II (Tli RNase H Plus) (Takara Bio, Otsu, Japan). The thermal cycling conditions were as follows: 95 °C for 30 s, 40 cycles at 95 °C for 5 s and 60 °C for 34 s. Three technical replicates were analyzed for each sample. Non-template negative controls were included in each run to detect possible contamination or carryover. A series of gene-specific primers were designed for RT-qPCR using Primer 3 software (http://bioinfo.ut.ee/primer3-0.4.0/) (Supplementary Table 1). The specificity of RT-qPCR reaction products was established via electrophoresis on 1.0% agarose gels, followed by sequencing. Transcript levels of genes were quantified using the $2^{-\Delta\Delta CT}$ method[35]. α-tubulin[36] and *RpL32*[37] were used as the reference genes in *B. dorsalis* due to their expression stability. The housekeeping gene *AaS7* and *actin* were used as endogenous controls for *A. aegypti*[38]. The gene expression of each sample was normalized to that of controls (taken as 1).

### In vivo detection and measurement of reactive oxygen species

We assayed in vivo reactive oxygen species and $H_2O_2$ production in the midgut of *B. dorsalis* as a measure of ROS activity. ROS levels in vivo were detected with dihydroethidium (DHE, D7008; Sigma). Midguts of 15-day-old flies were dissected in PBS containing 2 mg/mL 3-amino-1, 2, 4-triazol, a catalase inhibitor (A8056, Sigma). Midgut sections were immediately incubated with 2 μM DHE in PBS at room temperature for 30 min in the dark, followed by successive incubation in 4% paraformaldehyde and 5% Triton X-100, respectively, for 30 min at room temperature. Fluorescence images were collected using an EVOS FL Auto microscope (Life Technologies, Carlsbad, CA, USA) at 10× magnification.

The $H_2O_2$ content was determined using a Hydrogen Peroxide Assay Kit (Beyotime Biotech, Shanghai, China)[15]. Twenty midguts were dissected for each biological replicate of *B. dorsalis*. Sixty midguts were dissected for each replicate of *A. aegypti*. Samples were homogenized in 200 μL lysis buffer (reagent kits available), centrifuged at $12,000 \times g$ at 4 °C for 10 min, and the supernatant collected. Aliquots of 50 μL supernatant and 100 μL test solution from the hydrogen peroxide assay kit were incubated for 30 min at 30 °C, followed by immediate detection at a wavelength of 560 nm using a microplate reader (BioTek Synergy H1, BioTek Instruments, Monte, USA). The $H_2O_2$ concentration was calculated according to a hydrogen peroxide standard curve. The measurement was repeated three times. Experiments were performed with three to five biological replicates.

For biogenic amine treatment, ten-day-old WT adult flies were reared in 10% sucrose solutions containing different concentrations of amines for 3 days prior to determination of changes in $H_2O_2$ content. Amine mixture including dopamine, tyramine, and serotonin was measured at 1 μM, 10 μM, and 100 μM. The effect of 10 μM serotonin, dopamine, and tyramine on the $H_2O_2$ level was detected, individually. All amines were purchased from Sigma-Aldrich (St Louis, MO, USA).

### dsRNA-mediated gene silencing

The detailed procedures for gene silencing in *B. dorsalis* are described elsewhere[13,15]. Specific dsRNA primers with the T7 RNA polymerase promoter at the 5' end were used to clone target sequence fragments via nested PCR (listed in Supplementary Table 1). PCR products of the coding region were used as the template for synthesis of dsRNA in vitro using the MEGAscript T7 transcription kit (Ambion, Austin, TX, USA). As a negative control, green fluorescent protein (GFP) dsRNA was biosynthesized. The quality and size of dsRNA products were verified via 1% agarose gel electrophoresis. The dsRNA concentrations were quantitated at 260 nm using a NanoDrop 2000 Spectrophotometer (Thermo Scientific, United States) and the products diluted with nuclease-free water to a final concentration of 4 μg/μL. RNAi experiments were performed by injecting 0.5 μL of 4 μg/μL solution of dsRNA into the thoracic hemocoel of ten-day-old *B. dorsalis* adults using a FemtoJet microinjection system (Eppendorf). Gene silencing experiments in *A. aegypti* were performed by injecting 1 μg of dsRNA into the thorax of cold-anesthetized ten-day-old females. The efficiency of dsRNA-mediated gene silencing was determined via RT-qPCR at 1–3 days after injection. Injected insects were allowed to recover under standard rearing conditions. Experiments were performed independently at least twice, with three to five biological replicates per experiment.

## P450 and GST activity assay

The p-nitrophenyl ether-O-demethylase activity of P450 in midgut was measured using p-nitroanisole as a substrate[39]. In total, 20 midguts from ten-day-old *B. dorsalis* adults were dissected in ice-cold PBS. For *A. aegypti*, one hundred midguts from ten-day-old mosquitos were dissected. Samples were homogenized with 200 μL of 0.1 M PBS buffer (pH 7.5) containing 1 mM ethylene diaminetetraacetic acid, 1 mM phenylmethanesulfonyl fluoride, 1 mM propylthiouracil, 1 mM dithiothreitol and 10% glycerol. The supernatant was centrifuged at 4 °C for 20 min at $12,000 \times g$ to obtain an enzyme source solution. A 20 μL aliquot of supernatant was added to PBS (0.1 M, pH 7.5) containing p-nitroanisole (0.2 μM) and NADPH (9.6 mM) for the enzyme reaction. After incubation at 30 °C for 30 min, absorbance at 405 nm was measured using a microplate reader (BioTek Synergy H1, BioTek Instruments, Monte, USA). P450 activity in the supernatant was calculated by creating a standard curve based on A405 values for different concentrations of p-nitrophenol. Protein concentrations in the supernatant were measured using the bovine serum albumin (BSA) assay kit (Sangon Biotech).

A GST activity assay kit (Sangon Biotech) was used to detect midgut GST activity. GST catalyzes the binding of glutathione to 1-chloro-2, 4-dinitrobenzene, which generates a product with an optical absorption peak at 340 nm. GST activity can thus be effectively calculated by measuring the rate of increase in absorbance at 340 nm. Midguts were dissected in ice-cold PBS, followed by homogenization with 200 μL reagent I (reagent kits available). Next, the supernatant was obtained by centrifugation at $12,000 \times g$ at 4 °C for 20 min. The enzymatic reaction was conducted following the manufacturer's instructions and GST activity of the supernatant calculated from the protein concentration.

## Toxicity assay

The toxicity of each insecticide to *B. dorsalis* strains was determined with the aid of sucrose solution incorporation bioassays. Gradient concentrations of each test insecticide were dissolved in 10% sucrose solution. Ten-day-old adults of each strain (same number of males and females) were treated with each concentration of insecticide. Before treatment with insecticide solution, flies were starved for 2 h. Mortality was recorded after 1 day for trichlorfon, tetraniliprole, deltamethrin, thiamethoxam, cyclaniliprole, fluralaner, spirotetramat, triflumezopyrim, emamectin-benzoate, and spinosad treatment. Mortality was recorded after 2 days for chlorpyrifos. For *A. aegypti*, ten-day-old females were starved for 2 h before treatment with insecticide solution. Gradient concentrations of beta-cypermethrin were dissolved in 8% sucrose solution. Mortality was recorded after 1 day for beta-cypermethrin treatment. Insects were scored as dead in cases of no response to gentle prodding.

Chlorpyrifos (purity ≥99.2%, CAS: 2921-88-2), tetraniliprole (purity ≥97.68%, CAS: 1229654-66-3), beta-cypermethrin (purity ≥98%, CAS: 1224510-29-5), deltamethrin (purity ≥99.6%, CAS: 52820-00-5), thiamethoxam (purity ≥97.68%, CAS: 153719-23-4), cyclaniliprole (purity ≥97.68%, CAS: 1031756-98-5) were purchased from Dr. Ehrenstorfer (Augsburg, Germany). Trichlorfon (purity ≥98.8%) was purchased from National Pesticide Quality Supervision and Inspection Center (Beijing, China). Fluralaner (purity ≥99%, CAS: 864731-61-3), and spirotetramat (purity ≥99.2%, CAS: 203313-25-1) were obtained from Shanghai Yuanye Biotechnology Co., Ltd (Shanghai, China). Triflumezopyrim (purity ≥99.8%, CAS: 1263133-33-0), and emamectin-benzoate (purity ≥97.6%, CAS: 137512-74-4) were purchased from Shanghai Anpu Cuishi Standard Technical Service Co., Ltd (Shanghai, China). Spinosad (purity ≥96.18%, CAS: 168316-95-8) was purchased from MedChemexpress Biotechnology (USA). Five to six concentrations of each insecticide were used to establish the log-probit lines. All treatments were conducted at a temperature of 25 ± 1 °C under a 16:8 h light:dark cycle. Experiments were performed with three biological replicate, each replicate containing 20 insects.

## Vitamin C treatment

Vitamin C (VC) is a known antioxidant. Based on previous findings[37], administration of VC to *B. dorsalis* was effective in reducing ROS in the gut.

Ten-day-old adults (same number of males and females) were treated with 100 mg/mL VC (Sigma) for 24 h and the gut bacterial load and ROS levels determined as described above. P450 and GST gene expression and enzyme activities were additionally detected. Experiments were performed with three to five biological replicates, each containing 20 midgut samples. The toxicity of trichlorfon to VC-treated flies was determined as described above.

## Transcription factor CncC–Maf binding site prediction

Promoter sequence information of detoxification genes was acquired using the *B. dorsalis* genome (https://i5k.nal.usda.gov/Bactrocera_dorsalis). The 2000 bp fragment upstream of the transcription start site of each gene was selected as potential promoter region for transcription factor binding site prediction. Transcription factors of the CncC–Maf binding site in the promoter regions were predicted with JASPAR[40].

## Dual-luciferase reporter gene assay

To validate the transcription factor CncC regulation of detoxification gene expression, a dual-luciferase reporter gene assay was conducted. The overexpression vector for *BdCncC* and the detoxification gene promoter vector were constructed according to Zeng et al.[41]. Specifically, the open reading frame sequence of *BdCncC* was amplified from *B. dorsalis* cDNA using primers in Supplementary Table 1, and homologous recombination cloning was performed using the BsmBI and Esp3I restriction endonuclease sites on the pcDNA3.1(+) plasmid. Additionally, ten P450 and GST genes were selected, and their promoter regions were amplified from *B. dorsalis* DNA using primers listed in Supplementary Table 1. Subsequently, the amplified promoter sequences were homologously recombined into the SacI and HindIII sites of the pGL3-Basic plasmid, which carries the firefly luciferase reporter gene. The internal reference plasmid used carried the Renilla luciferase reporter gene under the control of the TK promoter (pGL4.74-TK).

HEK-293T cells (Beyotime Biotech) were seeded evenly in a 48-well plate and cultured for 24 h (reaching approximately 85% confluency), followed by plasmid transfection using Lipofectamine 3000 (L300015, Invitrogen). The transfection system consisted of a 1:1 ratio of pcDNA3.1(+) BdCncC to pGL3-Basic-promoter, with pGL4.74-TK constituting one-tenth of the total system. The treatment group received pcDNA3.1(+) BdCncC+pGL3-Basic-promoter+pGL4.74-TK, while the control group received the pcDNA3.1+pGL3-Basic-promoter+pGL4.74-TK plasmids. After 48 h of transfection, luciferase activities of both firefly and Renilla were measured using the Dual-Luciferase Reporter Assay System (E1910, Promega), following the manufacturer's instructions for cell lysis and reagent addition. Subsequently, readings were obtained using the chemiluminescence module of a microplate reader (BioTek Synergy H1, BioTek Instruments, Monte, USA). The assay included four biological replicates, and values were recorded for subsequent analysis.

## LC-MS/MS

Monoamine concentrations in midgut were detected according to the report of Davla et al.[42]. For preparation of samples for HPLC-MS, midguts of 30 flies (10 days old) for each genotype were dissected and immersed in ice-cold PBS. We dissected midguts to avoid cuticle contamination because serotonin and dopamine are intermediates in the sclerotization of insects[12]. Dissected midguts were centrifuged and PBS removed. Samples were immediately homogenized in 0.1% formic acid solution. After centrifugation at $12,000 \times g$ for 20 min, the supernatant was collected. An aliquot (1 μL) of supernatant was diluted 15 times to determine the protein concentrations at 595 nm using BSA as a standard (Sangon Biotech). Samples were stored at −80 °C until LC-MS analysis. Preliminary analytical conditions were developed using reference standards in FA solution containing dopamine, serotonin, tyramine, octopamine, and melatonin.

Mixed standard and sample extraction solutions were quantified using ultra-high liquid chromatography tandem triple quadrupole mass spectrometry (UHPLC-QQQ/MS, 1290-6470, Agilent Technologies, CA, USA)

under the following conditions: HESI source voltage 3500–4000 V, VCharging 500–1000 V, gas flow 6 L/min, Sheath gas flow 10 L/min, Sheath gas heater 350, gas temperature 300 °C, and nebulizer 45 psi. Chromatography of monoamines was performed on an Agilent Eclipse Plus C-18 analytical column (50 mm × 2.1 mm ID × 1.8 μm particle) based on adaptation of a reported method[42] using gradient elution of a binary solvent system that incorporated A (100% water) and B (100% acetonitrile) as follows: 0–1 min, 5% B; 1–6 min, 80% B; 6–8 min, 80% B; 8–12 min, 5% B. A valve was used to divert the first 0.25 min to "off-line" waste to eliminate the high level of salts eluting in the solvent front from extracted samples. The solvent flow rate was 300 μL/min. A 5 μL injection volume was used. The MS acquisition time was 12.1 min—the total run time was 14.1 min. The chromatographic system was determined to be stable and reproducible by injecting a reference solution three times at the beginning and at the end of the chromatographic run from a single vial containing reference solution.

## Statistics and reproducibility

Mortality data were corrected using Abbott's formula[43] and $LC_{50}$ values (50% killing concentration of flies) and 95% fiducial limits of $LC_{50}$ for each strain examined with Probit analysis using SPSS (Statistical Package for the Social Sciences) 27.0 software. Two $LC_{50}$ values were considered significantly different if their 95% fiducial limits did not overlap[44]. Other statistical analyses were performed using Prism 8 (GraphPad software). The two-tailed *t*-test was used for unpaired comparisons between two groups of data. For comparison of three or more sets of data, one-way ANOVA was performed, followed by Tukey's multiple comparison test. Differences were considered statistically significant at *p* values < 0.05. Data are expressed as mean ± SEM. Details of all statistical analyses are presented in the corresponding figure legends. The original data can be found in the Supplementary Data file.

## Reporting summary

Further information on research design is available in the Nature Portfolio Reporting Summary linked to this article.

## Data availability

RNA-seq data have been deposited in NCBI GenBank with accession numbers PRJNA1100975. Source data are provided in Supplementary Data 1. Uncropped blot images are included in Supplementary Information (Supplementary Fig. 6).

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

## Acknowledgements

We thank Ding-xin Jiang (South China Agricultural University) for assistance with the mosquito experiments. This work was supported by National Key R&D Program of China (Grant No. 2021YFC2600400).

## Author contributions

Y.X.Q. designed the experiments, analyzed data, and wrote the manuscript. T.Z. performed the majority of the experiments and helped analyze data and write the manuscript. F.Y.T. and H.W. helped with RNA isolation, gene cloning, western blot, and qPCR. Y.J.X. and Y.Y.L. contributed experimental suggestions and strengthened the writing of the manuscript. All authors reviewed, critiqued and provided comments to the text.

## Competing interests

The authors declare no competing interests.
