## [Peer Review File · Communications Biology]

Reviewers' comments:

Reviewer #1 (Remarks to the Author):

Comments:

The authors provide many evidences that AANAT1 regulates the CncC/Maf pathway through the production of ROS. However, the change in the sensitiveness to insecticides in AANAT1 mutant is not so solid. This minor change might be caused by the physiological effects in AANAT1 mutant fly. The role of AANAT1 in insecticide resistance needs to be carefully checked before consideration for accept.

1. L49-L52, why was AANAT1 chose in this paper? The introduction that AANAT1 will play some roles in detoxification is not so confident. Many genes are highly expressed in insect gut, why was AANAT1 selected in this paper, but not other genes?
2. L67-69, Is the AANAT1 mutant sensitive to other kinds of insecticide? Since the AANAT1 mutant shows reduced activities of many GSTs and P450s which are involved in the detoxification metabolism, the AANAT1 mutant might be more sensitive to many pesticides compared with the wild type.
3. L72-73, Compared with the WT strain, LC50 values were significantly decreased for both chlorpyrifos (2.04-fold) and trichlorphon (2.91-fold) in the BdorAANAT1 knockout strain. In resistance monitoring, the resistance ration between 0 and 5 is believed to be still sensitive. In this paper, the LC50 values were decreased for about 2~3 fold in mutant fly, however, this change might be caused by the physiological effects. Since the reference shows that AANAT1 regulates cuticle pigmentation and ovary development of the adult oriental fruit fly (Wang et al., 2022), we may infer that loss of AANAT1 might also inhibit the gut development, leading to weaker fly which is more sensitive to insecticides.

In addition, though GSTs, P450s and UGTs play important role in detoxification, they still have other roles, such as regulating development.

4. L76, BdorAANAT1 transcripts were mainly detected in the midgut region (Figure 2A). Was BdorAANAT1 expressed in other tissues, such as fatbody?
5. L77-79, BdorAANAT1 immunoreactivity was not detected in midgut of the mutant strains (Figure 2B). From Figure 2A, it shows that the transcription level of BdorAANAT1 in midgut is about 28 fold that in foregut, the immunostaining of the foregut from the same gut could be presented as a control. To better present the data, the intact gut with foregut, midgut and hindgut are needed for the wild type and mutant.
6. L97, Figure 5F-K should be Figure 3F-K.
7. Figure 3 shows that antioxidant vitamin C almost play the similar role as loss function of

BdorAANAT1. Is the wild type fly with vitamin C sensitive to insecticides?

8. L679 Figure S5 should be Figure S3.

Reviewer #2 (Remarks to the Author):

Zeng et al. aim to investigate the role of arylalkylamine N-acetyltransferase 1 (AANAT1) in the regulation of detoxification enzyme expression in two insects, *Bactrocera dorsalis* and *Aedes aegypti*. They shown that knockout by CRISPR/Cas9 or Knockdown by RNAi of BdAANAT1 reduce the ROS content in the midgut that has for effect to reduce the expression of the two transcription factors involved in the expression of detoxification genes, CncC and Maf and lead to a decrease the activities of P450 and GST. The flies were more sensitive to insecticides. Same results were obtained on mosquito and the role of biogenic amides in the reduction of ROS content was demonstrated.

The results are original, well designed, the manuscript is well written and provides new insights in the field of regulation of detoxification gene expression, which is still too rarely investigated. I have only minor suggestions to improve the manuscript, please see below

Results

- Line 71, the authors should justify the choice of trichorphon, this molecule is used as an acaricide and not an insecticide, why this choice? This is not obvious.
- Lines 107 to 111, it was not clear why the authors wanted to study a resistant strain. No information on this strain was given. Is this strain characterized for the mechanism responsible for resistance? Was it known that resistance in this strain involves detoxification enzymes?

Discussion

- Lines 207 to 209, if AANAT1 is important for survival, is a particular phenotype observed in flies that are knockout for AANAT1?

Materials and methods

- Line 346, one housekeeping gene is not enough to perform qPCR on *Aedes aegypti*, please respect MIQE, which are the international recommended rules for publishing qPCR results. Have a look to:
 - Bustin et al., 2009 The MIQE guidelines: Minimum information for publication of quantitative real-time PCR experiments. DOI: 10.1373/clinchem.2008.112797
 - Bustin and Wittwer., 2017 MIQE: A Step Toward More Robust and Reproducible Quantitative PCR Clin Chem. DOI: 10.1373/clinchem.2016.268953
- Line 369, please give details on monoamines, the name of the company where compounds were purchased and the concentrations used. Same remark for lines 408 to 420, give this information too for insecticides.

Minor points:

- Line 56, remove “e” at the end of develop.
- Line 195, correct “heterodimerization » by heterodimerization.
- Line 372, please correct and give names of the first author for the references rather than a number (19, 22) because the reference list is classified by author name and not by number.
- Line 429, a reference is needed for genome or at least a website address or a Genbank number.
- Line 432, please correct JASPR by JASPAR and the reference is not correct because the authors used the first name rather than the surname of the authors, please correct Elodie by Portales-Casamar.

Dear Reviewers:

Thank you very much for the valuable comments concerning our manuscript entitled “AANAT1 regulates insect midgut detoxification through ROS/CncC pathway”. Those comments are all valuable and very helpful for revising and improving our paper, as well as the important guiding significance to our researches. As suggested, the sensitivity of AANAT1 mutants to nine more insecticides were detected. We examined AANAT1 expression across additional tissue types including fat body, and Malpighian tubules, and intact gut. We add one more housekeeping reference genes in our qPCR analysis. Based on the reviewers’ comments, we try our best to revise the manuscript. Moreover, using a dual-luciferase reporter gene assay, we further confirmed that overexpression of BdcncC significantly enhances the promoter activity of the 10 targeted P450 and GST genes (revised Figure 4D). The main corrections in the paper and the responses to the reviewer’s comments are as following:

Reviewer #1 (Remarks to the Author):

Comments:

The authors provide many evidences that AANAT1 regulates the CncC/Maf pathway through the production of ROS. However, the change in the sensitiveness to insecticides in AANAT1 mutant is not so solid. This minor change might be caused by the physiological effects in AANAT1 mutant fly. The role of AANAT1 in insecticide resistance needs to be carefully checked before consideration for accept.

1. L49-L52, why was AANAT1 chose in this paper? The introduction that AANAT1 will play some roles in detoxification is not so confident. Many genes are highly expressed in insect gut, why was AANAT1 selected in this paper, but not other genes?

Response: Thank you very much. Our previous study revealed that serotonin modulates *B. dorsalis* gut ROS level (Zeng et al., 2022). AANAT1 is well known for its role in serotonin metabolism. Considering AANAT1 is highly expressed in insect gut (Hintermann et al., 1996, Wang et al., 2022), and ROS is involved in the transcription of detoxification-related genes (Palli, 2020), we propose that AANAT1 plays a regulatory roles in gut to modulate insect detoxification process. These information can be found in the introduction section (L38-L44).

2. L67-69, Is the AANAT1 mutant sensitive to other kinds of insecticide? Since the AANAT1 mutant shows reduced activities of many GSTs and P450s which are involved in the detoxification metabolism, the AANAT1 mutant might be more sensitive to many pesticides compared with the wild type.

Response: Thank you very much for the valuable advice. We detected nine other insecticides. Compared to WT, significant decreases in LC₅₀ values of *AANAT1*^{10/10} strain were observed for chlorpyrifos (2.04 fold), trichlorfon (2.91 fold), deltamethrin (1.88 fold), cyclaniliprole (1.72 fold), triflumezopyrim (1.74 fold), fluralaner (2.20 fold), spinosad (3.94 fold), emamectin-benzoate (2.05 fold), tetraniliprole (3.01 fold), and spirotetramat (1.99 fold) (Table S2 and Figure 1C). However, there was no significant difference in the LC₅₀ value of thiamethoxam between the two strains. These indicate AANAT1 regulated genes may be not involved in the detoxification process of thiamethoxam.

3. L72-73, Compared with the WT strain, LC50 values were significantly decreased for both chlorpyrifos (2.04-fold) and trichlorophon (2.91-fold) in the *BdorAANAT1* knockout strain. In resistance monitoring, the resistance ration between 0 and 5 is believed to be still sensitive. In this paper, the LC50 values were decreased for about 2~3 fold in mutant fly, however, this change might be caused by the physiological effects. Since the reference shows that AANAT1 regulates cuticle pigmentation and ovary development of the adult oriental fruit fly (Wang et al., 2022), we may infer that loss of AANAT1 might also inhibit the gut development, leading to weaker fly which is more sensitive to insecticides.

In addition, though GSTs, P450s and UGTs play important role in detoxification, they still have other roles, such as regulating development.

Response: Thank you very much for the valuable advice. Knockout of *BdorAANAT1* gene did not significantly affect the length of the foregut, midgut, and hindgut (Figure S3A). Compared with WT, the midgut width of *AANAT1*^{10/10} mutants did not change significantly (Figure S3B). Based on the literature, I think it's much more difficult to increase folds in susceptibility than in resistance (Wang et al., Nat Commun. 2018 Nov 16; 9(1):4820).

4. L76, *BdorAANAT1* transcripts were mainly detected in the midgut region (Figure 2A). Was *BdorAANAT1* expressed in other tissues, such as fatbody?

Response: Thank you very much. We detected weak fluorescence signals of *BdorAANAT1*-positive cells in Malpighian tubules (Figure S1A and B). *BdorAANAT1* immunoreactivity was not detected in the fat body (Figure S1C and D).

5. L77-79, *BdorAANAT1* immunoreactivity was not detected in midgut of the mutant strains (Figure 2B). From Figure 2A, it shows that the transcription level of *BdorAANAT1* in midgut is about 28 fold that in foregut, the immunostaining of the foregut from the same gut could be presented as a control. To better present the data, the intact gut with foregut, midgut and hindgut are needed for the wild type and mutant.

Response: Thank you very much. We revised according to your suggestions. The intact gut with foregut, midgut and hindgut are detected for the wild type and mutant (revised Figure 2B).

6. L97, Figure 5F-K should be Figure 3F-K.

Response: Thank you very much. We revised according to your suggestions.

7. Figure 3 shows that antioxidant vitamin C almost play the similar role as loss function of *BdorAANAT1*. Is the wild type fly with vitamin C sensitive to insecticides?

Response: Thank you very much. Yes, the wild type flies with vitamin C treatment are more sensitive to insecticide. Compared to control, significant decreases in LC₅₀ value of vitamin C-treated flies was observed for trichlorfon (5.07 fold) (Figure 3L).

8. L679 Figure S5 should be Figure S3.

Response: Thank you very much. We revised according to your suggestions.

Reviewer #2 (Remarks to the Author):

Zeng et al. aim to investigate the role of arylalkylamine N-acetyltransferase 1 (AANAT1) in the regulation of detoxification enzyme expression in two insects, *Bactrocera dorsalis* and *Aedes aegypti*. They show that knockout by CRISPR/Cas9 or Knockdown by RNAi of BdAANAT1 reduce the ROS content in the midgut that has for effect to reduce the expression of the two transcription factors involved in the expression of detoxification genes, CncC and Maf and lead to a decrease the activities of P450 and GST. The flies were more sensitive to insecticides. Same results were obtained on mosquito and the role of biogenic amides in the reduction of ROS content was demonstrated.

The results are original, well designed, the manuscript is well written and provides new insights in the field of regulation of detoxification gene expression, which is still too rarely investigated. I have only minor suggestions to improve the manuscript, please see below

Results

- Line 71, the authors should justify the choice of trichlorfon, this molecule is used as an acaricide and not an insecticide, why this choice? This is not obvious.

Response : Thank you very much. Trichlorfon is an irreversible organophosphate acetylcholinesterase inhibitor, and is a selective insecticide. Trichlorfon is toxic to target insects through direct applications and via ingestion. This insecticide is widely used to control insects, including *B. dorsalis* (Müller et al., 1989, *Angew Parasitol*; Cheng et al., 2017, *Microbiome*; Lin et al., 2013, *J Econ Entomol.*). This molecule is also used as an acaricide.

- Lines 107 to 111, it was not clear why the authors wanted to study a resistant strain. No information on this strain was given. Is this strain characterized for the mechanism responsible for resistance? Was it known that resistance in this strain involves detoxification enzymes?

Response: Thank you very much. To detect the role of AANAT1 in insecticide resistance, we study a resistant strain. Previous study of our group on this resistant strain showed that the gut symbiont *Citrobacter* sp. (CF-BD) is responsible for resistance (Cheng et al., 2017, *Microbiome*). Moreover, compared to sensitive strain, the expression level of detoxification genes in our study including LOC105233810, LOC105230935, LOC105225363, LOC105227474, LOC105228201, LOC105225817, LOC115066421, LOC105222104 were all significantly increased in trichlorfon resistance strain (RNA-seq data, our group unpublished data). Based on the published paper, detoxification enzymes are involved in *B. dorsalis* resistance to trichlorfon.

1) *Sci Rep.* 2018 Jul 25;8(1):11223. doi: 10.1038/s41598-018-29622-0.

2) *J Econ Entomol.* 2004 Oct;97(5):1682-8. doi: 10.1603/0022-0493-97.5.1682.

Discussion

- Lines 207 to 209, if AANAT1 is important for survival, is a particular phenotype observed in flies that are knockout for AANAT1?

Response: Thank you very much. AANAT1 regulates cuticle pigmentation and ovary development of the adult oriental fruit fly (Wang et al., 2022). Compared with the WT, the cuticle color of AANAT1 knockout adults was much darker. Moreover, the female AANAT1 knockout mutant had a smaller ovary than the WT, and laid far fewer eggs.

Materials and methods

- Line 346, one housekeeping gene is not enough to perform qPCR on *Aedes aegypti*, please respect MIQE, which are the international recommended rules for publishing qPCR results. Have a look to:

- Bustin et al., 2009 The MIQE guidelines: Minimum information for publication of quantitative real-time PCR experiments. DOI: 10.1373/clinchem.2008.112797

- Bustin and Wittwer., 2017 MIQE: A Step Toward More Robust and Reproducible Quantitative PCR Clin Chem. DOI: 10.1373/clinchem.2016.268953

Response: Thank you very much for the valuable suggestions. Two housekeeping genes, including *actin* and *As 17*, were used in the revised manuscript.

- Line 369, please give details on monoamines, the name of the company where compounds were purchased and the concentrations used. Same remark for lines 408 to 420, give this information too for insecticides.

Response: Thank you very much. We revised according to your suggestions.

Minor points:

- Line 56, remove “e” at the end of develop.

Response: Thank you very much. We revised according to your suggestions.

- Line 195, correct “heterodimerization » by heterodimerization.

Response: Thank you very much. We revised according to your suggestions.

- Line 372, please correct and give names of the first author for the references rather than a number (19, 22) because the reference list is classified by author name and not by number.

Response: Thank you very much. We revised according to your suggestions.

- Line 429, a reference is needed for genome or at least a website address or a Genbank number.

Response: Thank you very much. We provided a website address for the genome in the revised manuscript.

- Line 432, please correct JASPR by JASPAR and the reference is not correct because the authors used the first name rather than the surname of the authors, please correct Elodie by Portales-Casamar.

Response: Thank you very much. We revised according to your suggestions.

REVIEWERS' COMMENTS:

Reviewer #1 (Remarks to the Author):

The authors provide additional evidences that AANAT1 regulates the CncC/Maf pathway through the production of ROS. These evidences are more confident. The evidence that knockout of BdorAANAT1 gene did not significantly affect the length of the foregut, midgut, and hindgut could preliminarily explain that knockout of BdorAANAT1 gene could not affect the development of gut. To better illustrate this issue, the cell proliferation in the gut from the BdorAANAT1 knockout strain are more convincing. The DAPI staining cells can be used for cell number statistics. The BrdU antibody staining or PH3 staining can be used to check the cell proliferation.

Reviewer #2 (Remarks to the Author):

The authors have made all the changes requested by the reviewers, the manuscript is suitable for publication.

Dear Editor and Reviewers:

Thank you very much for the valuable comments. The main corrections in the paper and the responses to the reviewer's comments are as following:

REVIEWERS' COMMENTS:

Reviewer #1 (Remarks to the Author):

The authors provide additional evidences that AANAT1 regulates the CncC/Maf pathway through the production of ROS. These evidences are more confident. The evidence that knockout of *BdorAANAT1* gene did not significantly affect the length of the foregut, midgut, and hindgut could preliminarily explain that knockout of *BdorAANAT1* gene could not affect the development of gut. To better illustrate this issue, the cell proliferation in the gut from the *BdorAANAT1* knockout strain are more convincing. The DAPI staining cells can be used for cell number statistics. The BrdU antibody staining or PH3 staining can be used to check the cell proliferation.

Response: Thank you very much for the valuable advice. We performed the experiments according to your suggestions. No significant difference was observed in the number of phospho-histone H3 (PH3)-positive cells and 4',6-diamidino-2-phenylindole (DAPI)-positive cells in the midgut between WT and *AANAT1*^{10/10} strain (Supplementary Fig. 3c-f). These results indicate that cell proliferation was not significantly affected in the midgut of *BdorAANAT1* knockout flies.